# The user and non-user perspective: Experiences of office workers with long-term access to sit-stand workstations

Lidewij R. Renaud[1]*, Erwin M. Speklé[1,2], Allard J. van der Beek[1], Hidde P. van der Ploeg[1], H. Roeline Pasman[1], Maaike A. Huysmans[1]

1 Department of Public and Occupational Health, Amsterdam Public Health Research Institute, Amsterdam UMC, Vrije Universiteit Amsterdam, Amsterdam, The Netherlands, 2 Arbo Unie, Occupational Health Service, Utrecht, The Netherlands

* l.renaud@amsterdammumc.nl

## Abstract

### Objective

Sit-stand workstations have been shown to be effective in reducing sitting time in office workers. The aim of this study was to explore reasons for use and non-use of sit-stand workstations and strategies to decrease sitting and increase physical activity in the workplace from perspectives of users and non-users, as well as from managers and ergo-coaches.

### Methods

Six group interviews with employees who have had access to sit-stand workstations for several years were conducted in a large semi-governmental organisation in the Netherlands. Verbatim transcripts were analysed using thematic analysis. Open coding was conducted by three researchers and codes and themes were discussed within the research team.

### Results

Thematic analysis resulted in two major themes: 1) Reasons for use and non-use and 2) Strategies to increase standing and physical activity in the workplace. Shared and distinct reasons for use and non-use were identified between users and non-users of the sit-stand workstations. The most important reasons for use indicated by users were that they had experiencing immediate benefits, including staying alert and increasing focus; these benefits were not acknowledged by non-users. Non-users indicated that sitting was comfortable for them and that they were therefore not motivated to use the standing option. Strategies to increase the use of the standing option included an introductory phase to become familiar with working while standing and to experience the immediate benefits that come from using the standing option. Furthermore, providing reminders to use the standing option was suggested as a strategy to increase and sustain the use of sit-stand workstations. Increased use may lead to a change in the sitting culture within the organisation, as more employees would adopt active movement behaviours.

**Data Availability Statement:** Data for this research consists of transcripts of group interviews, which contain identifying information. Participants in this

study did not consent to make their data publicly available and in line with privacy regulations, publication is forbidden by our data protection officer at Amsterdam UMC. Anonymized selections from the transcripts can be made available upon request, to qualified researchers. Such requests can be addressed to SPH@vumc.nl.

**Funding:** One of the authors (ES) is affiliated with a commercial occupational health service (ArboUnie). The funder provided support in the form of salaries for the author [ES], but did not have any additional role in the study design, data collection and analysis, decision to publish, or preparation of the manuscript. The specific role of this author is articulated in the 'author contributions' section.

**Competing interests:** We would like to up-date our competing interest statement as follows: One of the authors (ES) is affiliated with a commercial occupational health service (ArboUnie). ArboUnie provided time to work on this manuscript and paid the salary of this coauthor (ES). This does not alter our adherence to PLOS ONE policies on sharing data and materials. Furthermore, the authors have declared that no competing interests exists.

## Conclusion

Immediate benefits of the use of the standing option–only mentioned by the users–was the most distinct reason to use sit-stand workstations. Future research should explore how to motivate potential users to adhere to an introductory phase in order to experience these immediate benefits, whether it is linked to the use of sit-stand workstations or other interventions to reduce sitting time.

## 1. Introduction

Recommendations within guidelines for physical activity include the general advice to sit less throughout the day [1–4], as prolonged sitting has been associated with several health risks. These health risks include type 2 diabetes, cardiovascular disease, and cancer [5–7]. Furthermore, a recent meta-analysis (only including studies with device-measured sitting time) showed associations between prolonged sitting time and increased risks of all-cause mortality [8]. Among office workers prolonged sitting is highly prevalent [9, 10], with an average of almost 5.5 hours per day of sitting at the workplace [11, 12], accumulating to total sitting times of above ten hours per day [13, 14]. Although some controversy exists around the health risks associated with prolonged sitting [15], interventions to reduce sedentary behaviour at the office, including the use of sit-stand workstations, have been increasingly implemented and extensively evaluated [16]. From a public health perspective, the rise in popularity of these interventions may be promising due to their strong potential to reduce time spent sitting among office workers, who have shown to be overall highly sedentary [17].

Recent studies exploring the use of sit-stand workstations found that they resulted in reduced sitting time compared to traditional sitting workstations. Long-term evaluations (six to twelve months) showed pooled reductions of 57 minutes per day in occupational sitting time [18]. Randomised controlled trials, such as the Stand-Up Victoria study, have been conducted in large office populations, introducing multi-component interventions—combining, for example, coaching reminders and sit-stand workstations. These evaluations have shown similar reductions of around 45 minutes occupational sitting time per day at 12 months follow-up [19, 20]. Furthermore, sit-stand workstations do not appear to negatively affect work performance, as opposed to cycling or treadmill desks which may [21]. Individual office workers seem to use many different strategies to reduce their sitting time and replace it with standing or physical activity. In the Stand-Up Victoria study, 82 separate strategies were identified by participants [22]. In another study, the behaviour change wheel [23] was used to develop a tailored intervention to reduce sitting, which identified 39 behaviour change techniques [24] aimed at providing information, rewards, and prompts for reducing sitting time [25]. To increase the quality and thus the reach of interventions, it is important to understand why some office workers chose to adopt the use of the standing option of their sit-stand workstations and why others do not, as well as whether other strategies are being used to reduce sitting. To examine this, qualitative research must be conducted which aims to explore and better understand these situations or behaviours [26].

A recent synthesis of qualitative research [27] included studies among sitting-based office workers which focussed on associated health problems [28] and possible strategies for reducing occupational sitting [29, 30]. These studies among sitting-based office workers found that workers could potentially be motivated to reduce their sitting time when provided with information which raised their awareness about the health risks of prolonged sitting [28, 29].

According to these workers, the most important motivation to reduce sitting time was to enhance musculoskeletal health rather than to reduce long-term health risks, such as diabetes or cardiovascular disease [28]. The synthesis also included qualitative studies conducted shortly after the implementation of sit-stand workstations and, focussing on user-experiences with the furniture [31–33], identified support from co-workers and managers to be an overall facilitator. A predominant sitting culture, meanwhile, was determined to be an overall barrier to reducing occupational sitting time [27]. However, participants in these studies did not have experience with interventions such as sit-stand workstations, or sit-stand workstations were implemented shortly before evaluations. The novelty of the furniture induced curiosity [31], increasing the risk for selection bias by only including participants who were enthusiastic about the furniture. This may have resulted in a more positive response towards sit-stand workstations than would be applicable to the whole population [33], by neglecting the perspective of office workers who did not use the furniture.

A recent study, which included interviews with current and previous users of sit-stand workstations, showed that sustained use of the furniture was dependent on personal considerations, such as the selection of a specific work task while using the standing option [34]. Although the inclusion of previous users added an additional perspective, previous users had all specifically requested to have a sit-stand workstation in the past, indicating similar levels of enthusiasm as the current users. Other perspectives under study included views of managers and occupational health practitioners, which showed that they were well-informed about the health risks of sitting [35]. To our knowledge, the perspective of non-users of sit-stand workstations, who have had access to the furniture at their workplace, without requesting it in the past, has not been included in previous studies. Were the non-user profile to be included, a deeper understanding of both reasons for use and non-use of sit-stand workstations could be established.

We previously conducted a survey study about the use of sit-stand workstations within a large office population (n = 1098) with long-term access to the furniture. In this study, we identified three user profiles of sit-stand workstations: daily users, monthly/weekly users, and non-users [36]. Users and non-users from the previous survey study were selected and invited to participate in the present qualitative study. The aim of the current study was to explore the reasons for use and non-use of sit-stand workstations, as well as to explore other strategies to decrease sitting and increase physical activity in the workplace. We incorporated different perspectives, including the perspective of users and non-users of sit-stand workstations and those of managers and employees who were trained as internal ergo-coaches. Insights, gained from taking a broader perspective, may be helpful for the development of tailored, wide-reaching interventions that reduce sitting time among office populations.

## 2. Methods

### 2.1 Study design and setting

This study is reported in accordance with the consolidated criteria for reporting qualitative research (COREQ) [37] (see S1 Table). Office workers from a large international semi-governmental organisation in the Netherlands were included. All employees at the worksite of this organisation (over 2000 in total) had had access to electrically adjustable sit-stand workstations since 1999 when the furniture was installed. Employees had personal, large workstations with a small, centred compartment adjustable to standing height. The study was conducted at the organisation's location during working hours. In total, six group interviews were held with between two and nine participants between April and June 2018. Each interview lasted about one and a half hours.

The Medical Ethical Committee of the Amsterdam University Medical Centres (located at VUmc) approved this study (2016.346). All participants gave written informed consent before commencing with the interviews.

## 2.2 Sampling

Participants were selected based on the answers they provided in the earlier survey study [36]. Potential participants were eligible for inclusion if they: 1) had a user profile (self-reported average use of the sit-stand workstation of at least once per week to several times per day) or a non-user profile (self-reported average use of the sit-stand workstation of less than once per week), 2) indicated interest in participating in further research, and 3) provided their email address. Participants were employees involved in the primary work process (involving computer work tasks) of the organisation, supportive personnel, managers, or internal ergo-coaches. In total, 144 employees were selected from the survey study and invited by email. The invitees included 70 users, 47 non-users, and 27 managers. Per user-profile, two group interviews were planned, one with employees involved in the primary work process and one with supportive personnel. Managers were invited as separate group (independent of frequency of use of the sit-stand workstation). Additionally, employees with a specific training for internal ergo-coach were invited to attend a group interview (n = 8). Internal ergo-coaches are employees who are specially trained in the ergonomic principles of a healthy workplace and who can be consulted by their colleagues about the ergonomic set-up of their workstations.

If potential participants agreed to participate, they were asked to respond to the invitation email and answer additional questions related to eligibility (frequency of use, job profile, availability) and individual characteristics (gender and age). In the selection of participants for the groups with specific user and job profiles, we aimed to maximize variation in participant characteristics (such as gender and age) however we did not exclude any employee from participation.

## 2.3 Research team

All members of the research team were involved in the development of a semi-structured interview guide. The main researcher (LR, female) conducted the group interviews, assisted by a research assistant who moderated (DS, see acknowledgments). Some participants who were involved in the primary work process of the organisation were familiar with the researcher (LR) because of their participation in another small quantitative project (n = 49), for which they were also invited based on their answers in the survey study [36]. Some members of the research team had current and extensive (ES) or past and minor (MH and LR) work experience as occupational ergonomists.

## 2.4 Data collection

The semi-structured interview guide was developed by the research team (see S1 File). The lead researcher (LR) provided (sub-)questions based on qualitative literature to date [29, 31, 32], which were set in the definitive format after debate within the research team on relevance and suitability for the target population. The interview guide included questions regarding reasons for use of sit-stand workstations, reasons for non-use of sit-stand workstations, and other actions taken to reduce sitting or increase physical activity (during and outside work). For managers, the guide also included questions about the importance of reducing sitting for employees, and the managers' potential role in this. For internal ergo-coaches, additional questions included main reasons to advise employees to use/increase use of the sit-stand workstation. Interviews were conducted in English and were audio recorded and transcribed verbatim

(by DS). Every group interview started with a warm-up exercise, in which participants paired off to discuss and note down reasons for use and non-use of the sit-stand workstations. The purpose of the warm-up exercise was to allow participants to become familiar with the topic and the group. After the exercise, reasons for use and non-use were discussed with the whole group, after which the group interview further evolved.

Based on field notes made by the moderator (DS), a one-page summary of each group interview was made (by LR) and sent to the participants. Comments or corrections from participants on these summaries were then processed. The comments indicated minor changes only and did not change the content of the summaries.

## 2.5 Data analysis

Data was analysed using ATLAS.ti 8 software and by three data coders (LR, MH and ES). After data collection, the six steps of thematic content analysis were used [38] in the process of open coding, to identify, review, and define (major) themes. Phase one involved reading the summaries and transcripts in their entirety to become familiar with the data. Phase two involved open coding of the four user and non-user interview transcripts (completed by LR) to generate initial codes. For one interview transcript (non-user profile, supportive personnel), open coding was also conducted by MH. Open coding of the management interview transcript was conducted by both ES and LR. During phase three, which encompassed searching for themes, general codes between LR and both coders (MH and ES) were discussed. When differences occurred, consensus between the coders was sought, which resulted in an initial coding tree. Additional codes from LR's coding of the transcripts were discussed with MH and consensus was again sought. In phases four and five, which encompassed reviewing and naming themes, the initial coding tree was reviewed by LR and HRP, and codes were reordered into potential themes. As a result, one additional separate theme—"Contextual and individual factors"—with subthemes and main codes was identified at this stage. Furthermore, two major themes (including themes, subthemes and main codes,) were identified: 1) Reasons for use and non-use of the standing option and 2) Strategies to increase the use of the standing option and physical activity at the workplace. This was again discussed within the whole research team and the hierarchy of the additional separate theme and the two major themes were set into thematic maps. Phase six included reporting on the data in the results section of this manuscript. Saturation seemed to only be established for one of the two identified major themes; no new codes which considerably altered the coding tree were identified during coding of the final transcript, although this was subject to the interpretation of the researchers [39].

## 3. Results

Within the first major theme–Reasons for use and non-use of the standing option–the following themes were identified: "Reasons for non-use", "Reasons for both use and non-use", and "Reasons for use". Reasons could result from personal preferences (i.e. conscious choice) or from external causes (i.e. unconscious motives). Within the second major theme–Strategies to increase the use of the standing option and physical activity–the following themes were identified: "Solutions addressing reasons for non-use of the standing option" and "Other strategies for being active at work." Reasons for use and non-use were thoroughly discussed during the interviews, and the perspectives of the managers and internal ergo-coaches seemed similar to those of the user. Still, their perspectives differed at some points, e.g. when they reflected on their roles as professionals (as described in section 3.5).

### 3.1 Participants

In Table 1, an overview of characteristics is provided for users, non-users, managers, and internal ergo-coaches. After invitation, 63 participants showed an interest in participating in the study, out of whom 31 ultimately were able to attend one of the six group interviews (which ranged in size from two to nine participants). Participants in the group interviews had a mean age of 49.1 (± = 5.4) years and were predominantly male (65%). In general, participants were highly educated long-term employees at the organisation who worked full-time. In the non-user groups, participants used the standing option never or less than once per month (except for one individual, who indicated using it less than once per week). In the user groups, participants used the standing option on a daily basis or at least several times per week. Internal ergo-coaches and managers used the standing option on a daily or weekly basis, with only one manager using it less than once per month or never.

### 3.2 Additional separate theme: Contextual and individual factors

The thematic map with subthemes and main codes of the additional separate theme "Contextual and individual factors" are presented in Fig 1, which included: cultural context, organisational context, interpersonal context, and individual attitudes and behaviour. These subthemes provided a deeper understanding into the background of the population within this organisation and as a whole regarding the concepts of sedentary behaviour and physical activity. There did not seem to be major differences in contextual and individual factors mentioned by users and non-users.

**3.2.1 Cultural, organisational, and interpersonal context.** Participants–who were from several European countries–acknowledged the cycling culture (e.g. using a bicycle for transportation to and from work) present in Dutch society. Furthermore, participants recognised that in society as well as in the organisation there was a predominant culture of sitting. An example mentioned was the culture during group meetings, in which the traditional furniture set-up addressed and facilitated sitting.

*"Culturally (think about chairing the meeting), you need people to comfortably sit and follow you. If you sit you follow. If you stand up after a while you start to talk, you lose your interest. So culturally this is the proof that [. . .] sitting is the comfortable position."*

**Table 1. Characteristics of 1) non-users, 2) users, and 3) internal ergo-coaches and managers.**

| Characteristics | Non-users of the sit-stand workstation | Users of the sit-stand workstation | Managers and Internal Ergo-coaches |
|---|---|---|---|
| Participated, N (N males) | 9 (6) | 11 (8) | 11 (6) |
| Mean age (SD) | 50.2 (5.1) | 46.9 (4.6) | 50.3 (5.8) |
| Work experience, mean years (SD)[1] | 18.6 (6.4) | 12.9 (6.4) | 21.1 (8.9) |
| Hours worked per week, mean (SD)[1] | 41.1 (3.8) | 37.5 (6.8) | 36.8 (12.1) |
| *Educational level[1]* | | | |
| PhD degree | 0% | 30% | 25% |
| Master's degree | 78% | 70% | 25% |
| Bachelor / post- secondary training | 22% | 0% | 50% |
| *Frequency of use of the standing option (N)* | | | |
| More than once per day | 0 | 7 | 7 |
| Once per month—4 times per week | 1 | 4 | 3 |
| Less than once per month / never | 8 | 0 | 1 |

[1] Data extracted from the survey study Renaud et al. [36].

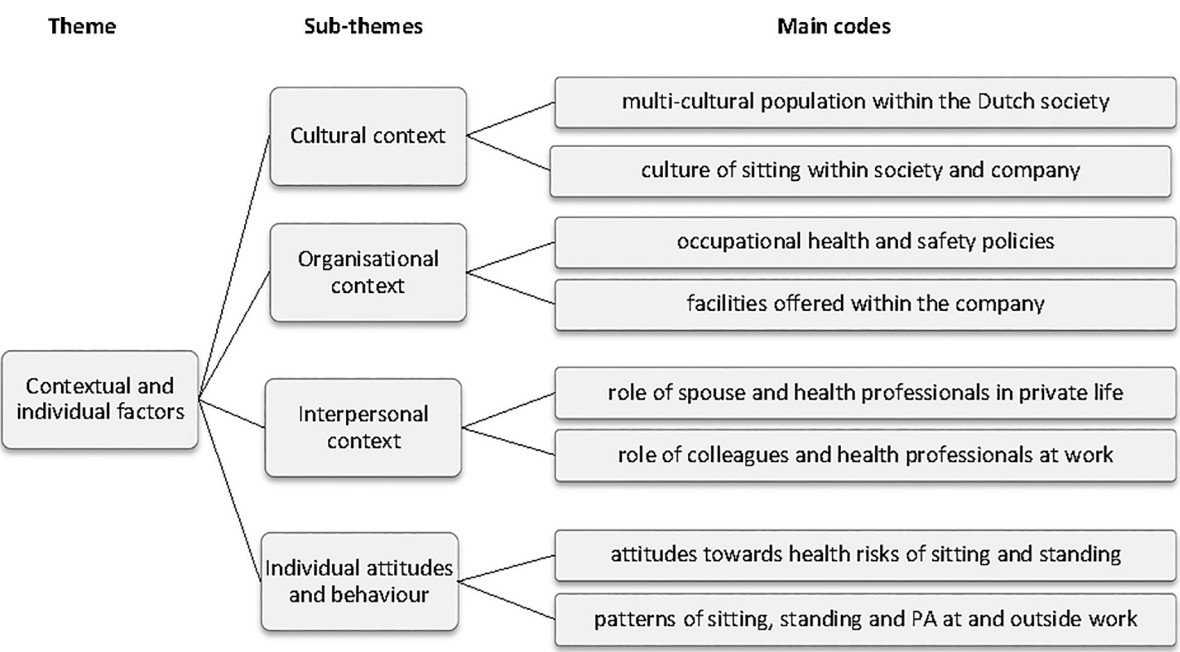

**Fig 1. Contextual and individual factors including subthemes and main codes.** PA = physical activity.

Non-user of the sit-stand workstation–supportive personnel

The organisational context offered information about occupational health and safety policies applied by the internal health and safety department, which included appointing and educating internal ergo-coaches. Furthermore, occupational health support was offered to the employees, ranging from computer rest-break software to numerous sports facilities. Within the interpersonal context, information was provided on the influence of others for being physically active. For example, colleagues were mentioned as an influence on (health) perceptions and behaviour at the workplace.

> *"It [the rest-break software] is very annoying because it stops me when I am in the middle of my work, but a colleague told me once 'but that is when you do not pay attention that you are sitting crooked and that is when you run into trouble afterwards.' And then I feel it when it is too late. And since my colleague told me that, I pay more attention to it and if the software tells me to stop or to slow down I actually do."*

Non-user of the sit-stand workstation–involved in primary work process

**3.2.2 Individual attitudes and behaviour.** Individual attitudes and behaviours included personal patterns of sitting, standing, and physical activity as well as attitudes towards the health risks of prolonged sitting and standing. Users reported varied frequency and duration of use of the standing option. Some indicated that they used it with a clear pattern when switching from standing to sitting, while others used it randomly, when they felt like it, without a specific pattern. Participants mentioned that durations of standing episodes lasted from between 15 and 30 minutes to up to several hours.

Participants from all group interviews indicated that they engaged in some sort of physical activity or other leisure activities outside the work place. Furthermore, all participants seemed

to be well-informed about the health risks associated with prolonged sitting. However, it was also noted by both users and non-users that being active could compensate for a sedentary lifestyle. Furthermore, the health risks of prolonged standing were not mentioned in any group interview.

> *"I think it is quite this kind of common message. Everybody sits too much, everybody eats too much. And we should get more active and standing is more active than sitting."*

User of the sit-stand workstation–involved in primary work process

### 3.3 Major theme: Reasons for use and non-use of the standing option

The thematic map of themes, subthemes, and main codes within the major theme Reasons for use and non-use of the standing option are shown in Fig 2. The subthemes within "Reasons for non-use" included habitual reasons and practical workplace issues. The theme "Reasons for both use and non-use" included the subthemes work content determines work posture, and experiences and behaviour. The theme "Reasons for use" included the subthemes health benefits and immediate benefits.

**3.3.1 Reasons for non-use.** *3.3.1a. Habitual reasons.* Among users of the sit-stand workstations a main reason for non-use seemed to be that a persistent habit of sitting prevented them from using the standing option more frequently. Furthermore, users indicated that they tended to forget to use the standing option.

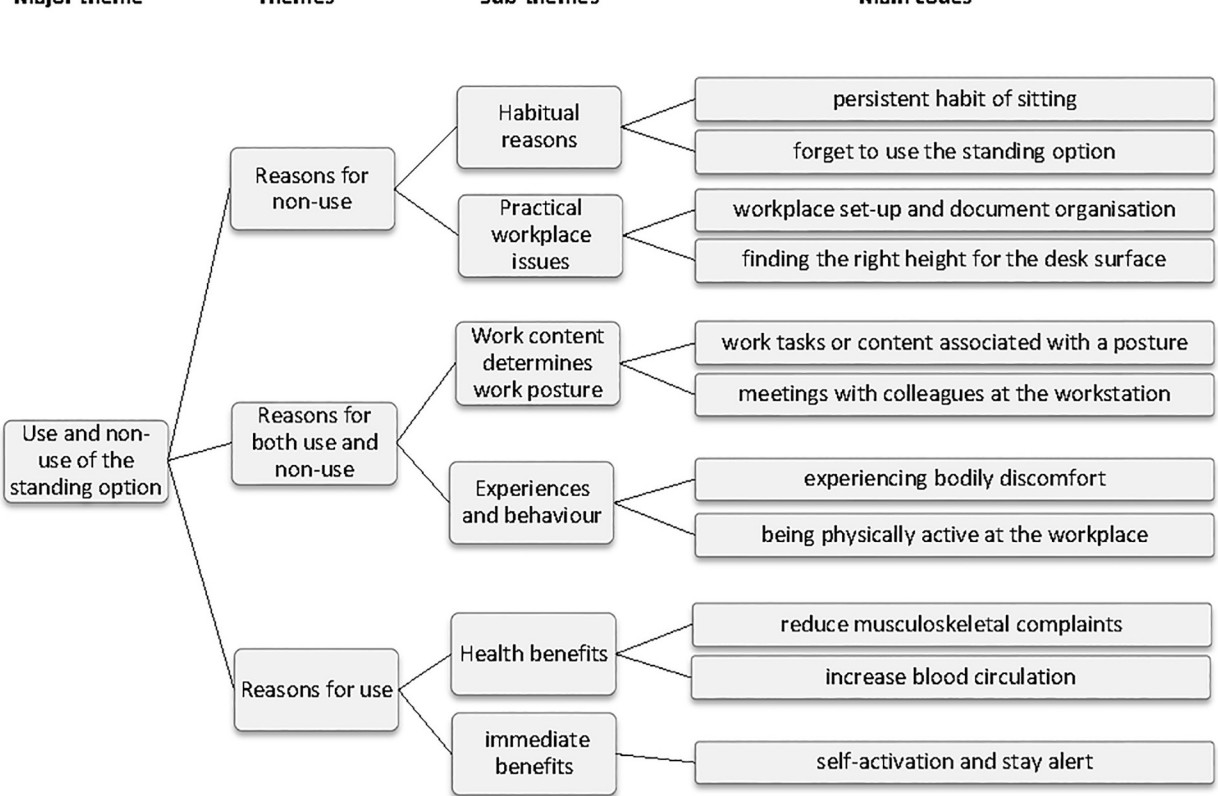

**Fig 2. Reasons for non-use and use of the standing option of sit-stand workstations including themes, subthemes and main codes.**

*"I do not use it every day but almost every day and sometimes I am so busy that I just forget about it and I am obsessed with other things."*

User of the sit-stand workstation–involved in primary work process

The persistent habit of sitting was also recognised by non-users but here neither motivation to change this habit nor advantages of using the standing option were acknowledged. Non-users also indicated that sitting is more comfortable, although some participants indicated they had never tried to work standing.

*"I find it comfortable working sitting, so I do not see the point myself to change behaviour. I never tried so that is also why I have not experienced the other side."*

Non-user of the sit-stand workstation–supportive personnel

*3.3.1b. Practical workplace issues.* Mostly, practical reasons for non-use were mentioned by non-users. These practicalities included the inability to put the workstation up because of poor (computer) cable management or because the desk was too full, since only a small part of the workstation surface is adjustable to standing height. The process of putting the workstation up, i.e. finding the right standing or sitting height, also takes time and was mentioned as another practical reason for non-use.

*"The problem is really [. . .] that when you put it up and you go down again, to get back to the position you had before where you were feeling comfortable. This is quite difficult"*

Non-user of the sit-stand workstation–involved in primary work process

**3.3.2 Reasons for both use and non-use.** *3.3.2a. Work content determines working posture.* Performing work tasks for which a high level of concentration is needed was mentioned as a reason for non-use, predominantly by users. On the other hand, changing to other work tasks were indicated as an incentive to use the standing option. Tasks that could be done while standing were indicated as easy, including checking emails or reading (from the screen). Both users and non-users indicated that tasks involving high volumes of typing or many paper documents were preferably done while being seated.

*"If it is routine work (which means we have to scroll through a lot of documents on the screen), it is easier when standing, but I find for typing and dealing with large dossiers it is easier to be sitting down where you can spread your work out."*

Manager–involved in primary work process

Non-users indicated that visiting colleagues at their personal workstation was a reason they preferred to be seated, while users indicated the same reason for putting the workstation into the standing position. The rationale behind this, for users, was that putting the workstation up allowed them to be at the same level as the visitor and shortened the meeting. Both user profiles suggested that team meetings were a reason to be seated, especially because of the traditional set-up of meeting rooms. Still, one-on-one or small group meetings in a standing position seemed to appeal to some users.

*"I only had a meeting maybe six weeks ago where we stood around, it was one of the few occasions. There was only five of us so you can stand around these smaller tables quite easily and*

*yes it actually brings people closer together 'cause you are leaning in and I think the dynamics of the conversation goes a little bit differently rather than sitting down in a line."*

Manager–involved in primary work process

*3.3.2b. Beliefs and experiences.* Avoiding or decreasing experiences of discomfort from being seated was indicated by users as a reason to use the standing option, as it motivated them to change their posture.

*"I get up when I cannot stay sitting in my chair anymore and I do not find a comfortable position anymore"*

User of the sit-stand workstation–involved in primary work process

On the other hand, after a period of standing, feeling discomfort and being tired were indicated as a reason for non-use and to return to a seated position. Experiencing discomfort from being in a standing position was also identified as a reason for non-use by non-users. However, this experience of discomfort was related to the inability to find a correct standing position at the workstation rather than the result of prolonged standing itself.

*"I mean it depends if I am walking around as well then it does not feel too tiring. But if you stay, if you stand all that time staring at or working with the screen it feels a bit... and then I feel myself leaning on the table. Not having the correct position so it is also not good."*

Non-user (previous user) of the sit-stand workstation–supportive personnel

From the user perspective, using the standing option could make it easier to move around at the office and to help them to be physically active.

*"I think with what I understood that movement is quite important and therefore, while standing, I can move more and it seems to help in my personal case so that is why I try to do."*

User of the sit-stand workstation–involved in primary work process

From a non-user perspective, being active at the workplace was mentioned as reason for non-use. This activity could be incorporated in work tasks (especially for supportive personnel who need to walk more), or incorporated deliberately by choosing active alternatives to sedentary activities.

*"I don't take the elevator; I take the stairs. And I rather have a coffee break after one and a half hour sitting than standing up at the desk. So, I consider that enough. So that is why I don't use the standing position."*

Non-user of the sit-stand workstation–involved in primary work process

**3.3.3 Reasons for use.** *3.3.3a. Health benefits.* Health benefits were mentioned by both users and non-users as reasons why it is (or would be) better to reduce sitting time and use the standing option. Better blood circulation was mentioned as a reason for use but without reference to reductions in long-term health risks such as cardiovascular disease.

*"Well it is the alternative for getting some movement and improving the blood circulation in your legs 'cause you get stiff from sitting. So, I feel it, I can feel it."*

User of the sit-stand workstation–supportive personnel

Experiencing or avoiding musculoskeletal complaints, such as low back pain, were indicated by users as a reason to start to use sit-stand workstations and by non-users as a potential reason to start using the sit-stand workstation.

*"If my back is killing me and standing would help, that would be probably an incentive to stand up every ten minutes."*

Non-user of the sit-stand workstation–involved in primary work process

*3.3.3b. Immediate benefits.* Experiencing immediate benefits were only mentioned by users of the sit-stand workstations. This reason to use the standing option included staying more alert (a way to activate oneself when feeling drowsy) and increasing focus during work (when a drop in concentration was experienced). Users explained that switching to a standing posture enabled them to switch their focus during work.

*"So, if I am sitting down, I have the tendency to. . . well, not immediately fall asleep but be more tired and less concentrated. And if I stand up I am more focussed and (if I am doing a search for instance), I can better keep the concentration for a longer time if I am standing up and moving a little bit."*

User of the sit-stand workstation–involved in primary work process

## 3.4 Major theme: Strategies to increase the use of the standing option and physical activity

The thematic map of the major theme Strategies to increase the use of the standing option and physical activity at the workplace (including themes, subthemes and main codes) is presented in Fig 3. The first theme included solutions "Addressing reasons for non-use of the standing option" with the subthemes of habitual reasons and practical issues. The other theme included solutions "Addressing other strategies", which included subthemes aimed at organisational, office environmental, and individual adjustments.

**3.4.1 Addressing reasons for non-use.** To address the habitual reasons for non-use, it was stated that the use of the standing option should be incorporated within daily routines. Several tips were provided by users to achieve this, mostly aimed at personal changes in habits. These included leaving the desk in an upright position when leaving the office and putting it in an upright position at fixed times. Also, slowly increasing the intervals of use were recognised as a factor for successful incorporation, with prolonged periods of absence from the office increasing risks of relapse into habitual sitting.

*"Every time you leave, raise the table. Because then you get the habit. And once you have the habit, then it is easy to follow because then you can just follow your body [. . .] The only problem I had was to restart after long holidays or after being really sick [. . .] then you sit more at some days and then you have to consciously switch it on again."*

Internal Ergo-coach

Reminders on a daily basis, for example from the rest-break software already available at personal computers, were mentioned as helping to maintain new or existing routines. Reminders timed on a less frequent or subconscious basis were also suggested, in order to

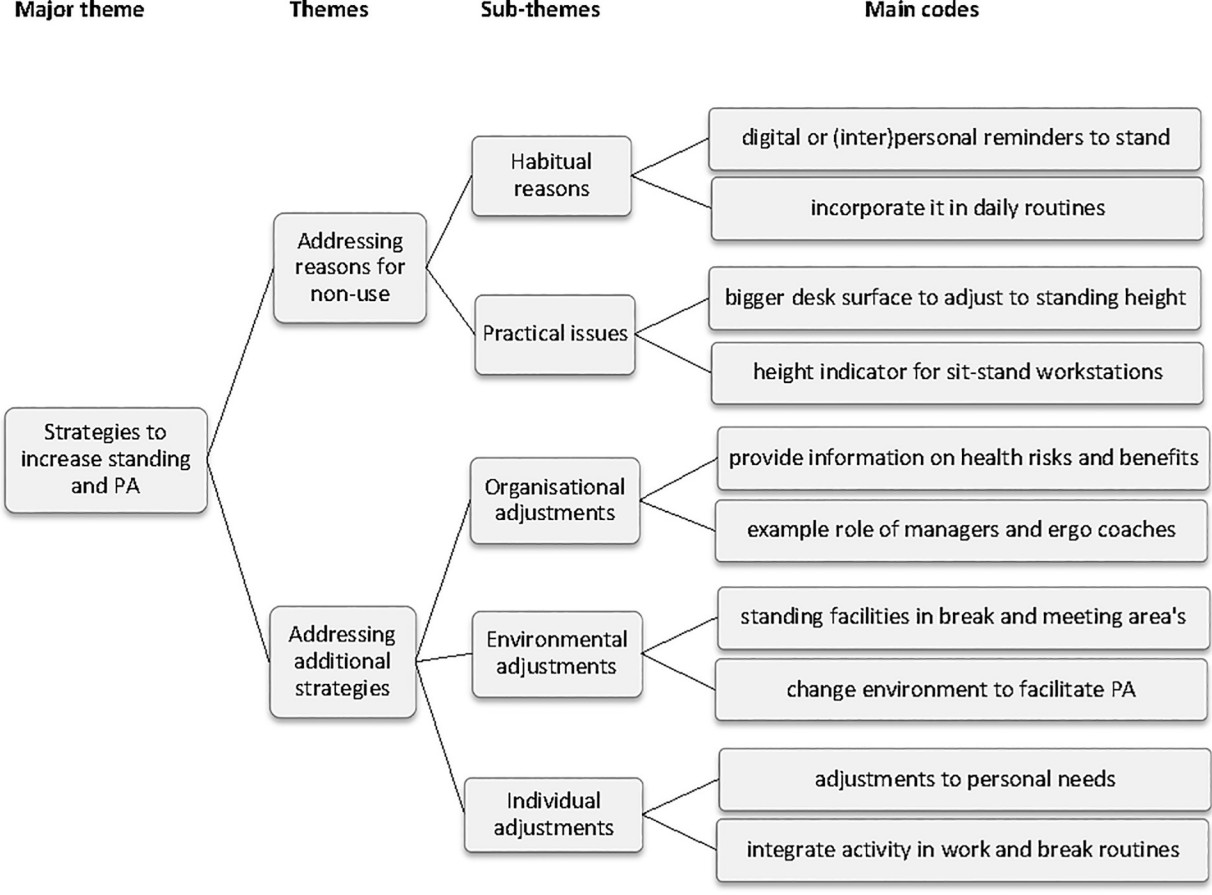

**Fig 3. Strategies to increase the use of the standing option and physical activity, including themes, subthemes and main codes.**
PA = physical activity.

reinforce the use of the standing option after a period of non-use. Examples included posters at the worksite or seeing a colleague who uses the standing option.

> *"It can be promptious [sic, a prompt] seeing a colleague with their desk up and so I think 'I have not done this for a while.'"*

Manager–involved in primary work process

Non-users mentioned addressing practical issues for non-use. Larger desk surfaces to adjust to standing height and digital height indicators to show the height of the desks, preferably including a memory function to put the workstation at a pre-set height, were suggested as solutions.

**3.4.2 Addressing additional strategies.** Within organisational adjustments, managers and internal ergo-coaches acknowledged that they could be role models for employees. For instance, the could serve as examples and provide information about the (health) benefits of using the standing option and being less sedentary at the office. Although the need for more information was also acknowledged by users and non-users, the role that managers and ergo-coaches could play in providing this information was minimally acknowledged. Environmental changes in the office building that were suggested included changes to facilitate physical activity, such as the implementation of desk bikes, treadmill desks, or standing aids (e.g.

standing stools). Other suggestions included the implementation of standing facilities in meeting rooms and the cafeteria. In these facilities, a partial replacement of sitting furniture was recommended, to offer a choice of whether to be seated or remain standing.

*"In a [meeting] room like this even, the set-up is for presentations but the back bit could have higher desks with this supportive standing so that some people can maybe opt to listen more actively. But then you do not have them, you know, one sitting one standing because then the view is a bit strange. But there is one area where people can choose to be more active in their position."*

Non-user of the sit-stand workstation–supportive personnel

Within the theme "Individual adjustments", several suggestions to integrate activity in work and break routines were provided in all group interviews. This included walking to colleagues instead of sending emails, taking the stairs or go for walks during lunch time. It seemed important to adjust interventions to personal needs. It was suggested that a facility room (i.e. a sort of library) could be set-up, where interested employees could try new tools such as desk bikes. Furthermore, it was emphasised again (by non-users) that sitting was comfortable for them and standing would not be an option but that sitting alternatives, such as active sitting chairs, could work.

*"But as I told you I was also looking for alternatives for sitting. So, there is a way, there is a problem, we are trying to find solutions, standing is not the one we like so we look for other solutions. That is, I think, the most important part of the game. There is always an alternative and we have to find it."*

Non-user of the sit-stand workstation–involved in primary work process

### 3.5 Manager and ergo-coach perspective

Managers and ergo-coaches were predominantly users of the sit-stand workstations, and they reflected similar reasons for use and non-use as well as similar strategies for increasing standing and physical activity as other users mentioned. However, other topics were also raised and discussed during group interviews with managers and ergo-coaches. Managers indicated that they could not only be examples for the employees in their departments by using the standing option, they could also play a role in normalising standing and fighting the sitting culture.

*"The people get used to, you know, if they go somewhere, a meeting or whatever, that you do them standing up. Because then it is not going to be weird to stand up. They will be more used to stand up because that would happen in other different set-ups, not only in the office but everywhere."*

Manager–involved in primary work process

Ergo-coaches also reflected on their role of serving as an example in using sit-stand workstations within their departments as well as in meetings with employees they visit for an ergonomic consult. They emphasised the importance of an introductory phase for potential users to familiarise themselves with standing. They indicated that this phase should last at least several days. After this, immediate benefits of using the standing option can be experienced and the use can be incorporated into daily routines. If this threshold of a trial period is not met, routine usage will fail, as was also revealed in a personal example.

*"At some point I said 'I want to do it for my health and because I cannot convince other people if I am not doing it myself.' And I wanted to test it also myself. [. . .] and then you sit for one hour and then it is already lunch time and then you missed it. But then after one week I forgot to think about it. I just realised 'oh I failed'."*

Internal Ergo-coach

## 4. Discussion

This study explored reasons for use and non-use of sit-stand workstations and strategies to decrease sitting and increase physical activity in the workplace. Within a population who had had access to sit-stand workstations for several years, we included perspectives of users and non-users, as well as managers and ergo-coaches. The most important finding was that users and non-users of the standing option indicated several similar reasons for use and non-use, in addition to other reasons that were users/non-users specific. Furthermore, some reasons seemed to be two-fold, both a reason for use and non-use at the same time. Subsequently, distinct and shared reasons and strategies for successfully using the standing option of sit-stand workstations suggest that different approaches to reduce sitting time in the workplace might apply for the two user profiles.

### 4.1 Distinct reasons between users and non-users

The most distinct reason between users and non-users seemed to be experiencing the immediate benefits of using the standing option, including staying alert and increasing focus, which was only mentioned by users. Users suggested that an introductory phase could be helpful for non-users to increase the use of the standing option. Still, long periods of absence from work could trigger relapse as indicated elsewhere [40].

Users indicated that they preferred to use the standing workstation when completing work tasks which did not require high levels of concentration, such as sending emails. Specific work tasks associated with standing have also been indicated in other studies [22, 25, 40]. Moreover, a recent study found that people did not identify "sitting" as an activity in and of itself, but rather thought of sitting in relation to the task they performed (e.g. computer work, monitor time, etc.). As such, the message "reduce sitting time" might be misinterpreted, as most sitting time could be linked to task-based behaviours [41].

After (long) periods of standing, users reported experiences of bodily discomfort in their lower extremities, which was a reason to return to a seated position. They seemed unaware of the risks associated with prolonged standing, such as varicose veins [42], with some using the standing option for several hours in a row.

Practical reasons for non-use, such as a desk surfaces being full or too small (since only part of the workstations could be put up), were mentioned mainly by the non-users. Such practicalities have previously been mentioned as reasons for non-use in populations with short-term access to sit-stand workstations [31, 32]. However, when the use of the standing option is part of daily routines, this no longer seemed to be a practical barrier, as only a few users mentioned this.

### 4.2 Shared reasons between users and non-users

Several reasons for use and non-use were shared by users and non-users. The need to focus during certain work tasks was a reason for users and non-users to work in a seated position. Although quantitative research has shown that work productivity might not be affected when sit-stand workstations are used [21, 43, 44], earlier qualitative research also noted perceived concerns about productivity while working in a standing position [29, 35, 45, 46]. In the

current study, statements about productivity (gains or losses) were not mentioned. Still, work tasks which required high focus (e.g. typing) were mentioned as preferably being done while sitting. This might indirectly mirror a decrease in perceived productivity if these tasks were done standing. In other studies there have been indications that high focus work tasks are prioritised over the use of the standing option [40], or that sitting postures are not consciously experienced during high-focus work tasks [25].

The perceptions of the cultural, organisational, and inter-personal context were similar in users and non-users. Similarities between the two user perspectives might be expected for the overall context, such as the sitting culture within society. Still, knowledge about health risks of prolonged sitting and levels of leisure (physical) activity were also equally addressed by users and non-users. This suggests that contextual and individual factors did not determine whether participants engaged in the use of the standing option or not.

In line with a broadly recognised sitting culture, both users and non-users mentioned that the persistent habit of sitting held them back from using the standing option more often. This was also found in other qualitative research [27], as well as in a quantitative study conducted in a large office population (N = 533) with sit-stand workstations [47]. Organisational culture has been identified as an important factor to successfully reducing sitting in office workers [48].

## 4.3 Recommendations for practice

Gardner et al. (2016) found that for interventions to reduce sitting, environmental restructuring and persuasion or education were the most promising behavioural change strategies [49]. When developing interventions including sit-stand workstations (environmental restructuring), adopting an introductory phase (persuasion) might lead to the immediate experience of benefits, which (as also indicated by the users in our study) could in turn result in long-term use. However, it was also indicated by users that this introductory phase takes time and energy and that persistence is needed for several days with increasing intensity of use. This intensity of use of the standing option might never be met by some office workers with a non-user profile. Non-users indicated that they feel comfortable when sitting and did not see any benefits in changing their routines. Non-adherence of this group might be inevitable [36] and other solutions, such as active office breaks incorporated into work tasks, might be more feasible for decreasing sedentary behaviour [50]. Still, an introductory phase to induce long-term use might also be worthwhile in such 'non-user' interventions.

It might also be useful to link sitting reduction efforts to specific office tasks, such as prompting standing by putting the sit-stand workstation up when leaving the office, as indicated by the users and internal ergo-coaches in this study. To increase adherence to interventions for reducing sitting time, frequent reminders could also be beneficial. Digital reminders (e.g. using specific sit-stand software) seemed effective in the long-term to increase the use of the sit-stand workstation at the population level [51, 52]. Also, seeing colleagues using the standing option was indicated as a reminder and prompt for use [25, 53]. This might become more evident as a transition is made from a sitting culture towards a more physically active work culture, in which organisational culture promotes healthy behaviours [25, 48]. Standing (group) meetings, including the proper furniture, might be implemented as a strategy to alter sitting norms and promote a more active culture. Still, it might be necessary to do more than just place standing tables in a meeting room to overcome sitting culture, as the use of these tables might also introduce physical and psychological discomfort during meetings [54], and an introductory phase might again be needed. Furthermore, when implementing interventions to reduce sitting time information about the health risks associated with prolonged standing

should also be provided [55, 56] by (for example) managers and internal ergo-coaches acting as role models.

Interventions to reduce sitting at the office should include components that address all employees, such as incorporating a more physically active work culture, as well as tailored components for employees with user and non-user profiles. Future research should look into how to identify employees who are potential users or non-users and whether differentiating between them could increase the reach and effectiveness of interventions.

## 4.4 Strengths and limitations

A strength of this study was that participants had had long-term access to sit-stand workstations, assuring that perceptions reflected routine behaviour and were not reflecting only on the novelty of the furniture. The variation in perspectives included was also a strength, with the non-user perspective being rather novel compared to recent literature. Furthermore, participants in our study had many different nationalities, which may contribute to higher generalisability and a broader perspective.

Limitations of this study included possible selection bias, because non-users were likely to be less enthusiastic about participating in a study about sit-stand workstations. The non-users we included in our study might therefore have been more enthusiastic and physically active, which may have resulted in similar contextual and individual factors between users and non-users. In the total office population of the organisation, there might be more variation in these factors between users and non-users, as indicated in earlier quantitative research [36]. Furthermore, generalisability to other office populations might be limited because this population was very highly educated, had private offices with personal sit-stand workstations, and performed specific computer work tasks as their primary work process. Still, similar outcomes have been established in other research concerning the user perspective. The non-user profile is very likely to be present in other (more general) office populations as well, yet this should be verified in future research. Because of the variation of perspectives, more information was provided from different angles and it was anticipated that it would be difficult to reach saturation for the two major themes. Still, Reasons for use and non-use seemed saturated when the coding of all transcripts was complete. This major theme was elaborately discussed from all perspectives, and no new codes that considerably altered the coding tree seemed to arise, although this was subject to the interpretation of the researchers [39]. Because of the clear research aim (to explore reasons for use and non-use of sit-stand workstations as well as to explore other strategies to decrease sitting and increase physical activity in the workplace), the coding process might have been influenced beforehand. However, all themes within the two major themes, and the additional separate theme, resulted from the data analysis process, suggesting that the influence was limited.

## 4.5 Conclusion

Users and non-users of sit-stand workstations identified shared and distinct reasons for their use and non-use of the standing option. Furthermore, they provided insight into potential strategies for increasing standing and physical activities at the workplace. The most important reason for users to use the standing option was the experience of immediate benefits, including staying alert and increasing focus. The contextual and individual factors, including the predominant sitting culture and personal attitudes towards the health risks of prolonged sitting, seemed important but were not perceived differently between users and non-users. When implementing interventions to reduce occupational sitting, such as sit-stand workstations, it could be helpful to include an introductory phase in order for potential users to experience

immediate benefits and become familiar with a new routine. Still, other strategies not involving sit-stand workstations might be needed to reduce sitting time in some office workers with the non-user profile. Future research should look into how to identify users and non-users before the implementation phase and how to motivate them to adopt an introductory phase, whether it is linked to the use of sit-stand workstations or other interventions to reduce sitting time.

## Supporting information

**S1 Table. COREQ checklist.**
(PDF)

**S1 File. Interview guide.**
(DOCX)

## Acknowledgments

We would like to acknowledge Dominique Stijnman (DS) for the valuable contributions to this study, including moderation and transcription of the interviews.

## Author Contributions

**Conceptualization:** Lidewij R. Renaud, Erwin M. Speklé, Allard J. van der Beek, Hidde P. van der Ploeg, Maaike A. Huysmans.

**Formal analysis:** Lidewij R. Renaud, Erwin M. Speklé, H. Roeline Pasman, Maaike A. Huysmans.

**Investigation:** Lidewij R. Renaud.

**Methodology:** Lidewij R. Renaud, Erwin M. Speklé, H. Roeline Pasman, Maaike A. Huysmans.

**Project administration:** Lidewij R. Renaud.

**Supervision:** Allard J. van der Beek, H. Roeline Pasman, Maaike A. Huysmans.

**Writing – original draft:** Lidewij R. Renaud.

**Writing – review & editing:** Lidewij R. Renaud, Erwin M. Speklé, Allard J. van der Beek, Hidde P. van der Ploeg, H. Roeline Pasman, Maaike A. Huysmans.

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
