## [Decision Letter · Decision Letter 0]

19 Feb 2020

PONE-D-20-01975

The user and non-user perspective: experiences of office workers with long-term access to sit-stand workstations

PLOS ONE

Dear Ms. Renaud,

Thank you for submitting your manuscript to PLOS ONE. After careful consideration, we feel that it has merit but does not fully meet PLOS ONE’s publication criteria as it currently stands. Therefore, we invite you to submit a revised version of the manuscript that addresses the points raised during the review process.

Please find reviews from two experts below. Overall, the reviewers are positive about your work but have some concerns about the coherency and clarity of the presentation of your manuscript, particularly with regards to the results section.  Please ensure all concerns are addressed in your resubmission. Please also provide information regarding the availability of the raw data from the manuscript.  

We would appreciate receiving your revised manuscript by Apr 04 2020 11:59PM. To enhance the reproducibility of your results, we recommend that if applicable you deposit your laboratory protocols in protocols.io, where a protocol can be assigned its own identifier (DOI) such that it can be cited independently in the future. For instructions see: http://journals.plos.org/plosone/s/submission-guidelines#loc-laboratory-protocols

We look forward to receiving your revised manuscript.

Kind regards,

Kathryn L. Weston, PhD

Academic Editor

PLOS ONE

Journal Requirements:

We note that one or more of the authors are employed by a commercial company: Arbo Unie.

Reviewers' comments:

Reviewer's Responses to Questions

**Comments to the Author**

1. Is the manuscript technically sound, and do the data support the conclusions?

Reviewer #1: Yes

Reviewer #2: Yes

2. Has the statistical analysis been performed appropriately and rigorously? 

Reviewer #1: N/A

Reviewer #2: N/A

3. Have the authors made all data underlying the findings in their manuscript fully available?

Reviewer #1: No

Reviewer #2: No

4. Is the manuscript presented in an intelligible fashion and written in standard English?

Reviewer #1: No

Reviewer #2: No

5. Review Comments to the Author

Reviewer #1: This paper needs reworking particularly on presentation of results and the English used within.

The authors also state that restrictions apply in terms of data availability #

Reviewer #2: Comments to the editor:

Thank you for the opportunity to review this interesting manuscript. This manuscript describes a qualitative study aiming to explore employee and management views of reasons for use and non-use of standing workstations and to explore views on strategies to decrease sedentary behaviour and increase physical activity. While I think this is an interesting study, I have several suggestions which I think should be addressed before the paper can be considered for publication. My specific comments are below.

Comments to the author:

I enjoyed reading this interesting study. I think the use of qualitative research methods is a novel approach which provides an in-depth insight into the views and opinions of employees using sit to stand workstations. However, I think that your manuscript could be strengthened by addressing my comments below.

Broad Comments

Introduction:

While the introduction cites important literature in the area of sit-to-stand workstations and sedentary behaviour, in places I felt that it lacked coherence and clarity. I think that in places the lack of coherence detracted from the overall justification for conducting the study. Overall, I think that the justification for conducting this study, using qualitative methods in particular, could be improved. Below is a list of questions I felt were unanswered by the introduction. Consideration of these questions could strengthen the justification for conducting the study:

• Why sit-to-stand workstations as opposed to other forms of sedentary behaviour interventions? Is there evidence to suggest that sit-to-stand workstations are more effective for reducing sedentary behaviour time/ improving health outcomes than other methods?

• Why are qualitative research methods the most appropriate data collection method for examining this specific research question?

• Why is this study novel? What gap is it addressing in the literature?

Methods:

I commend the authors for reporting their study in line with the COREQ guidance.

Although I think most aspects of the methods undertaken have been reporting well, based on the reporting of the thematic analysis conducted in this paper, I do not believe that a reader could follow the analysis steps undertaken, in a future study. To strengthen this section, the authors could report specifically how they undertook each of the six steps of thematic analysis, as outlined by Braun and Clark.

Results:

In places I found the results section difficult to follow. For example the following sentence (line 178): “Themes within reasons for use and non-use of the standing option were reasons for non-use, reasons for both use and non-use, and reasons for use”. I think this sentence is referring to sub-themes within the larger theme of “reasons for use and non-use of standing option”. I think that the clarity of the results section could be improved like so: Themes within “reasons for use and non-use of the standing option” were “reasons for non-use”, “reasons for both use and non-use”, and “reasons for use”. I think that there needs to be a clear differentiation throughout the results section, to denote where the names of themes or subthemes are referred to. I think that the results section should be updated throughout to reflect this suggestion.

Specific comments:

Introduction:

Page 3 line 48- although I do not speak or understand Dutch, it appears that Reference 8 is a secondary website quoting data from another study. Is this the most appropriate reference for this study? Has the data been published elsewhere? I apologise if this is incorrect. Were the participants from this study Dutch? If so I think this should be clarified here. Can the authors please clarify if “accumulated total sitting time” refers to all sedentary behaviour domains (e.g. sitting at home as well as at work), or just at work?

Page 3, line 48. As far as I can see, reference 11 does not explore the use of sit to stand work stations for reducing sedentary behaviour. Can the authors confirm if the correct reference has been cited here?

Page 4, line 53- please can the authors expanding upon what was included in “multi-component” interventions?

Page 4, line 54- Can the authors please clarify what they mean by “similar effects” here? Do they mean similar to a different type of intervention or similar to the 57 minute reduction in sedentary time reported in reference number 12.

Page 4, line 55- was there any evidence in references 12, 13 and 14 of the variation in reductions in sedentary behaviour? Perhaps standard deviations could be reported to support this claim.

Page 4, line 58- should this line read “between studies” rather than “between individuals”? Please could the authors clarify here whether there were differences in intervention strategies within an individual trial, or whether across a number of trials the strategies were different?

Page 4 line 58, please could the authors provide some more information about the “Stand-up Victoria” study, as it is not mentioned before now.

Second paragraph of the introduction- the authors have reported here a number of studies which examine a range of techniques used to modify sedentary behaviour. But, in my opinion, they have not fully justified why sit to stand work stations may be an appropriate option for reducing sedentary behaviour, and most importantly, whether this reduction in sedentary behaviour improves health or wellbeing outcomes. Please refer to my earlier major comment regarding the introduction of the paper, which I think will help to address this point.

Page 4 line 65- I commend the authors for their discussion of a recent meta-synthesis of qualitative work examining the acceptability and feasibility of sit to stand work stations. However, I felt this important paragraph lacked coherence in places, and I found it difficult to read. I would suggest that the authors restructure the paragraph so that the findings of review are clearly reported. Perhaps the types of studies should be described together and then the findings of the review should be stated, to improve the flow of this paragraph.

Page 4, line 81. Should this sentence perhaps read “previous and current users of sit to stand workstations” instead of “users and ceased users”?

Page 4, line 83- can the authors please clarify here what they mean by “the inclusion of ceased users was a novelty”. Do they mean a novelty in terms of the evidence base?

Page 4 line 85- can the authors please clarify here what they mean by “indicates parallels with the user profile”. What specifically does this indicate is similar between the “user” and “ceased user” groups in reference number 24?

Page 5, line 26. Reference number 26- could the findings of this study be described in more detail in the introduction?

Methods:

Can the authors please confirm how many participants were include in each focus group? I note this information has been included in the results section, but typically this information is reported in the methods section.

Page 5, line 104- Can the authors report here the number of staff within the participating organisation? This will give an indication of the percentage of total staff that were recruited to the study.

Page 5, line 108- I think “group interviews” are typically referred to as focus groups in qualitative literature. I would suggest the authors amend this throughout the manuscript.

Page 7, line 144- please can the authors clarify what they mean by “deliberate debate” which was conducted during focus group schedule development and what this process consisted of. Apologies, I am unfamiliar with the term.

Page 7, line 152- “This warm-up exercise was intended to get familiar with the topic and the group”, can the authors please clarify here if this sentence is referring to the participants themselves, or the researcher?

Page 7, line 157- the authors state here that “The summaries were used to predefine the major themes for data analysis”. Was this an evidence based analysis decision? If so, please could the authors cite an appropriate source here? If not, could the authors consider in the limitations section of the discussion if pre-defining themes based on a summary document given to participants could have inhibited the analysis process as outlined by Braun & Clarke?

Page 8 first paragraph. The data analysis process that were undertaken and reported here seem to contradict the process which was reported on line 157. At what point during data analysis was the coding tree developed? Was it developed from the summary sheet given to participants or following thematic analysis process?

Results

Page 8, Line 177- sentence beginning “Furthermore, an extra theme”. Is this sentence complete? It does not make sense as it stands.

Page 8, line 179-80- The sub-theme “reasons for use” is expanded upon here but the other subthemes are not. Is there a rationale for this choice?

Page 8- does the first paragraph of the results section present a summary of the main findings? If so, I think naming this section accordingly would guide the reader more clearly through the results section. Furthermore, the structure of the results section does not match this summary section (the sections are in a different order in the summary and the text). To help the flow of the results section, the summary section should list the themes in the order they are presented in the text, or the result section should be reorganised accordingly.

Page 9- would the participants section and Table 1 not fit better within the methods section?

Page 9, table 1- does the row “Work experience” refer to the length of time the participants have been employed by the organisation where the research was conducted? Or total work experience in any organisation? Please could the authors clarify this in the table by either making the heading clearer, or adding a legend?

Page 9, line 193-194- please can the author clarify what they mean by “hardly ever used”? I think this needs to be more specific.

Page 10, line 225: While interesting, I am not sure that discussion of children being a barrier to leisure time physical activity behaviour is relevant to the research question. I think this should be removed.

Page 11- line 244-45: Can the authors present a quote here to illustrate the participants views about the “health risks” associated with prolonged sitting.

Page 15, Line 343- Please can the authors expanding upon the “several health benefits” reported by participants as reasons for use of the sit to stand work stations? Perhaps an illustrative quote here would help.

Page 16, line 360: please can the authors clarify what they mean by “self-activation”. Is this a term from the literature? I think a definition is required here.

Page 17 line 378- please can the authors confirm who the “several tips” were provided by? Were they provided by the researchers to the participants, or from participant to participant?

Discussion

Page 22, line 499: Please can the authors clarify what they mean by “barely mentioned” here.

Page 23 line 521: can the authors cite any empirical evidence here reporting the “acute benefits” of reducing sedentary behaviour. I think this statement needs to be more specific, and also supported by the evidence base.

Page 23, line 533: can the authors confirm what they mean by “physically dynamic work culture”. I have not read this phrase elsewhere. This also needs to be clarified on line 540.

Page 23, line 535: If reference number 46 finds that standing meeting furniture causes physical and psychological discomfort, why would they be recommend for use here? What alternatives would the authors suggest to facilitate “standing meetings”?

Page 24, line 550. The authors state here that the inclusion of a range of nationalities within the sample is a strength of their study. While I agree, I cannot find information presented in the manuscript about the nationalities of those included. I think this information could be included in or near to Table 1.

Conclusion

Page 25, line 578: How do the authors propose that researchers identify users and non-users of an intervention before an intervention is implemented?

6. PLOS authors have the option to publish the peer review history of their article (what does this mean?). If published, this will include your full peer review and any attached files.

Reviewer #1: No

Reviewer #2: No

---

## [Author Response · Author response to Decision Letter 0]

20 Mar 2020

1.we have adjusted the manuscript style and changed the file naming

2a and b. Please see our response in the letter to the editor and the attached statement from the data protection officer

3.We have adjusted the affiliations

4a. and b. Please see our cover letter for our updated competing interest statement

---

## [Decision Letter · Decision Letter 1]

26 Mar 2020

PONE-D-20-01975R1

The user and non-user perspective: experiences of office workers with long-term access to sit-stand workstations

PLOS ONE

Dear Ms. Renaud,

Thank you for submitting your manuscript to PLOS ONE. After careful consideration, we feel that it has merit but does not fully meet PLOS ONE’s publication criteria as it currently stands. Therefore, we invite you to submit a revised version of the manuscript that addresses the points raised during the review process.

While both reviewers agree your paper is much improved following the first revision, they still share major concerns about the 'readability' of the paper and strongly suggest the use of less complicated terminology and reworking the grammar and syntax through the manuscript. As you will see below, Reviewer 1 has provided an extensive and thorough list of queries and concerns, which must be fully addressed in your revision.  Please also ensure to address the remaining issues highlighted by Reviewer 2. 

We would appreciate receiving your revised manuscript by May 10 2020 11:59PM. To enhance the reproducibility of your results, we recommend that if applicable you deposit your laboratory protocols in protocols.io, where a protocol can be assigned its own identifier (DOI) such that it can be cited independently in the future. For instructions see: http://journals.plos.org/plosone/s/submission-guidelines#loc-laboratory-protocols

We look forward to receiving your revised manuscript.

Kind regards,

Kathryn L. Weston, PhD

Academic Editor

PLOS ONE

Reviewers' comments:

Reviewer's Responses to Questions

**Comments to the Author**

1. If the authors have adequately addressed your comments raised in a previous round of review and you feel that this manuscript is now acceptable for publication, you may indicate that here to bypass the “Comments to the Author” section, enter your conflict of interest statement in the “Confidential to Editor” section, and submit your "Accept" recommendation.

Reviewer #1: (No Response)

Reviewer #2: All comments have been addressed

2. Is the manuscript technically sound, and do the data support the conclusions?

Reviewer #1: Yes

Reviewer #2: Yes

3. Has the statistical analysis been performed appropriately and rigorously? 

Reviewer #1: N/A

Reviewer #2: Yes

4. Have the authors made all data underlying the findings in their manuscript fully available?

Reviewer #1: No

Reviewer #2: No

5. Is the manuscript presented in an intelligible fashion and written in standard English?

Reviewer #1: No

Reviewer #2: Yes

6. Review Comments to the Author

Reviewer #1: Thanks again for the opportunity to re-review this paper. As I mentioned previously, I think this has a lot of potential. Although a lot of progress has been made to improving the standard of the paper, I would again suggest the authors resubmit a revised version. My main concerns are still in relation to the language used which was confusing and made the paper difficult to read and follow. I wouldn’t be confident it would be written clearly enough to be accessible to non-specialists. I found the language complicated and sentence structure clunky and difficult to navigate. Please consider reworking the grammar, syntax and choose less complicated terminology. Once this paper is more “reader friendly”, I think it will make a good addition to the literature. I do not feel this is presented in an intelligible fashion and written in standard English in some places. I am not clear on a number of items which make the interpretation of the results difficult. For example, I find the familiarisation and practise concepts difficult to understand without more information. A

Line 17: May benefit from insertion of country where the organisation is.

Line 18: From what I can see, open and axial coding is mentioned in this line and then not mentioned again in the manuscript. Please either further explain open and axial coding in the main text or consider removing from abstract.

Line 23: Is the term self-activation the best to use here? Please consider an alternative as it is not very clear what is meant. The term “activation” draws my mind to psychological theories.

Line 26: Consider clarification of practise periods and familiarisation here and throughout the paper. Will come back to this point when I meet it in the main body.

Line 28: Again will touch on this in main body, but more clarity around kinds of information is required here and main body.

Line 40: “INCLUDING SOLELY” please rephrase. Perhaps “only including”.

Line 40: I understand I mentioned this before and the advice was taken on board, but the English here isn’t the best. Perhaps use “device measured” in this instant to allow better flow. My apologies for confusion.

Line 42: 5.5 hours…. Per day? Please insert unit.

Line 46: If it has already been extensively evaluated, why is more research being done by you?

Line 46/47: Is this gain in popularity referring to the desks or the interest in occupational sitting levels etc.?

Line 48: Please be careful with terms here as inactive doesn’t mean sedentary, necessarily. Which is it? Or both?

Line 50: “INDEED”, is this a necessary word?

Line 57: Consider treadmill desk here.

Line 57: Is the word remarkable required here. In my opinion, here it doesn’t add to the scientific style of writing.

Line 57/58: Confusing sentence, unsure what is meant. Please rephrase for clarity.

Line 59/60: Please use the correct term “behaviour change wheel” and reference it (Michie).

Line 61: Again, correct term please. “Behaviour change techniques”. Please also reference.

Line 64/65: I don’t believe this is a solid rationale for why the qualitative methods were used. Please review.

Line 69-71: Please rewrite for clarity.

Line 89: Is the “at least in the past” at the end of this sentence needed, or is it repetition as mentioned earlier in the sentence?

Line 91-94: is this a recommendation for future research or a limitation of the study? Not sure how this fits here or the point being made. Please clarify.

Line 96-98: Please clarify here the link between the 2 studies more clearly. That one followed from the other, same research team, offices etc.

Line 99: Which study is being referred to?

Line 110: what is this referring to (±2000) plus or minus 2000? Employees? Please clarify in particular what the ± symbols is indicating here.

Line 112: Are all these desks then 21 years old? Are they dated? Or have the desks been upgraded?

Line 114: Please provide more precise information on the number of participants and the time of each session. Consider using descriptive stats for further clarity.

Line 124: Is this all that applied? How were they selected? Is this based on the inclusion/exclusion criteria?

Line 127-128: Please review the English used.

Line 130: Frequency of what?

Line 130-131: Why was collaboration required? Is it necessary to report this?

Line 134: What kind of consultation was available about the organisation?

Line 136: Why is gender an eligibility criteria?

Line 138: So you excluded some people based on their gender? Please explain

Line 140-142: Is this a requirement of the journal? I am not convinced this is critical to the paper if not required by journal specifications.

Line 147: Is this repetition?

Line 148-149: Unsure why this is relevant here.

Line 153: How was debate resolved?

Line 157: Why use word internal here? were there external too?

Line 167: Did you get much correction/comments, is this mentioned in results?

Line 176: I mentioned this earlier, here open coding only is referred to?

Line 182-183: Why was saturation not required, please include a reference for this.

Line 184-185: Was this only evident here?

Line 204: Please reconsider the use of the term attendant here.

Line 227-228: Why is the cycling culture relevant to occupational sitting?

Line 238: Is software a “facility”?

Line 244-248: Why so many quotation marks used here? Are they different quotes?

Line 253-256: This is an example of the hard to read sentences. Consider rephrasing and using randomly instead of random.

Line 256: How was this measured?

Line 317-318: How does the height impact conversation dynamic, please explain.

Line 335-337: I do not understand what is meant here.

Line 342-343: Please rephrase this sentence especially, “making them more prompted”.

Line 347: Is work place a better term here?

Line 365-367: hard to follow this sentence. Please reword.

Line 374: Please consider changing self-activation as mentioned previously.

Line 376: Does direction of switch matter here?

Line 386-387: Please consider only using the word strategy once in this sentence.

Line 391-392: As above, use/user is used 3 times. Please reword.

Line 393: Please consider using “in an upright position”, rather than “up”.

Line 394 (2nd half): Is there a quote for increasing intervals?

Line 395: Is this an absence of use or physical absence from the office?

Line 402: What is the reason for the brackets here?

Line 406: Is promptious a typo? I can’t find it in a dictionary.

Line 406-407: Why so many quotation marks here also?

Line 410: What is a digital height indicator?

Line 411-412: Why so?

Line 419: provide information on (benefits of) using the standing option- is this relating to the desk use or just in general?

Line 419-420: What kind of information?

Line 420-421: Their role in doing what exactly?

Line 433: Is this double “or” required?

Line 433-435: Is this 2 points in the one sentence? The first half seems not to fit with the second half?

Line 436 “Active sitting”, what does this mean here? does the quote reflect this?

Line 444-446: please reword, what are the similar dialogues? from where?

Line 453-458: Could be made more concise? A lot of repetition here. What kind of threshold of acquaintance? is this the best term to use? i am not sure how a familiarisation period would work? Please explain what is meant here? 

Line 471: “for users or non-users”, would and fit better here?

Line 472 and throughout: Is distinctive the best term? Consider something like "different"?

Line 479: “Practice period”, is this linking back to the acquaintance threshold etc.? Please ensure consistent terminology to avoid confusion.

Line 484-485: What does this sentence add? it might be worth summarising for readers the kinds of tasks if this adds to the paper. 

Line 489-490: Was back not mentioned too?

Line 490: “They seemed hardly aware”- Please rephrase.

Line 496: How was the integration explored in this study? What does it mean?

Line 497: Please be more specific, little or none?

Line 502: Can you really say this definitively? Perhaps the word suggested is better here?

Line 507-509: please reword to avoid confusion.

Line 513: Does this mean both groups had the same knowledge/concerns about prolonged sitting?

Line 516: UK or US spelling?

Line 517: Is it a reason for not using it? Is reason the best word here? is it more of a side effect or something along those lines?

Line 526: Why the brackets?

Line 527-529: Throughout, what is meant by this practise period isn’t clear to me. Is it more like an introductory phase?

Line 530-531: How can you say this if they haven’t changed anything?

Line 533: “the persuasion by practice periods”- Unclear what is meant here.

Line 537: Our study? Or this study?

Line 543: Including/included x 2, please rephrase.

Line 546: please clarify why the placing of a desk result in psychological discomfort?

Line 551-552: Why is this identification important? do you just mean by asking someone if they think they’d use? Is this before they received a desk? Please elaborate.

Line 562-563: I do not grasp what is meant here. Please clarify.

Line 565: In the workplace population you looked at or all over the world?

Line 568: Did each person have their own individual office and their own individual desk? Did I miss this earlier? Apologies if I did. Please consider the impact this may have particularly on role modelling of others given that you cannot see the others in the building.

Line 572-573: Would benefit from a reference here.

Line 577-578: Do you think this happened? What was in place, if anything, to stop this?

Line 582-584: Please consider splitting into 2 sentences.

Reviewer #2: The authors have attended to my comments well and I believe it has strengthened the paper greatly. I do agree with Reviewer One, who stated in their review that in places the manuscript was difficult to follow. While I believe this has been addressed in places, I have made a number of comments below indicating where there are some grammatical/ spelling/ formatting errors that when corrected will further improve the flow of the manuscript.

The Line numbers I have used refer to the document with tracked changes.

Throughout the manuscript, both UK and US English spelling are used. I would suggest referring to the journal guidelines about which form should be used, and amending throughout. For example line 516 recognized (US spelling) and line 523 organisational (UK spelling).

Line 51- I suggest rewording this sentence to “could reduce health risks associated with prolonged sitting in people who are inactive”.

Line 67- I am unclear from the description what the Stand Up Australia study actually entailed. I think further description is needed here (perhaps just a sentence about the study design/ methods).

Line 76- qualitative used twice- sentence needs to be rephrased.

Line 82- check grammar here

Line 98- typo “dependent of”

Line 117- should this read ergo-coaches?

Line 127- consider changing the word obtained to installed.

Line 153- Does this mean that some individuals volunteered and were eligible to participate but were not selected? Does this have any implications in terms of bias this may have introduced?

Line 171- if the phrase “based on literature to date” is to be used here, I would suggest a citation (or citations) to the pertinent literature is required here.

Line 180- check grammar here “get familiar with the topic”.

Line 185- what was involved in the “processing of comments and queries” were any changes made to the interview transcripts?

Line 192- check grammar “was done”

Line 188- While I commend the authors for including more detail in the data analysis section, I found it difficult to follow in parts. I am still unclear where each of the six steps for thematic analysis begin and end. Perhaps it would be useful to use phrases like “in the first/ second/ third stage of thematic analysis....” to guide the reader through this section.

Line 203: This sentence is quite unclear. I am unclear how a hierarchy of two major themes and an extra theme were presented in three conceptual frameworks. Is there a simpler and more concise way that this could be explained? Perhaps the authors could consider refer back to the language used by Braun and Clarke to guide the writing of this section?

Line 332- typo “indicated as reason”

Line 496- consider changing “looked into” to “explored”

Line 499- consider rephrasing “the most important finding seemed that”

Line 525- consider rephrasing “switch back” to “return”. Switch back is colloquial language.

Line 532- I think the phrase “none to little” needs to be changed here. I think the authors need to specifically state whether it was a few participants that mentioned this, or none at all, as it cannot be both.

Line 592- should this phrase be “physically active” rather than “physical active”?

Line 633- consider rephrasing “get familiarised” to “become familiarised”?

7. PLOS authors have the option to publish the peer review history of their article (what does this mean?). If published, this will include your full peer review and any attached files.

Reviewer #1: Yes: Aoife Stephenson

Reviewer #2: No

---

## [Author Response · Author response to Decision Letter 1]

13 Apr 2020

Please see our attached file: Response to reviewers

---

## [Decision Letter · Decision Letter 2]

28 Apr 2020

PONE-D-20-01975R2

The user and non-user perspective: experiences of office workers with long-term access to sit-stand workstations

PLOS ONE

Dear Ms. Renaud,

Thank you for submitting your manuscript to PLOS ONE. After careful consideration, we feel that it has merit but does not fully meet PLOS ONE’s publication criteria as it currently stands. Therefore, we invite you to submit a revised version of the manuscript that addresses the points raised during the review process.

As detailed below, both reviewers are much happier with the revised manuscript, but several queries remain about the language, structure and syntax throughout the manuscript. Please ensure all remaining queries are addressed in your revision.

We would appreciate receiving your revised manuscript by Jun 12 2020 11:59PM. To enhance the reproducibility of your results, we recommend that if applicable you deposit your laboratory protocols in protocols.io, where a protocol can be assigned its own identifier (DOI) such that it can be cited independently in the future. For instructions see: http://journals.plos.org/plosone/s/submission-guidelines#loc-laboratory-protocols

We look forward to receiving your revised manuscript.

Kind regards,

Kathryn L. Weston, PhD

Academic Editor

PLOS ONE

Reviewers' comments:

Reviewer's Responses to Questions

**Comments to the Author**

1. If the authors have adequately addressed your comments raised in a previous round of review and you feel that this manuscript is now acceptable for publication, you may indicate that here to bypass the “Comments to the Author” section, enter your conflict of interest statement in the “Confidential to Editor” section, and submit your "Accept" recommendation.

Reviewer #1: (No Response)

Reviewer #2: All comments have been addressed

2. Is the manuscript technically sound, and do the data support the conclusions?

Reviewer #1: Yes

Reviewer #2: Yes

3. Has the statistical analysis been performed appropriately and rigorously? 

Reviewer #1: N/A

Reviewer #2: N/A

4. Have the authors made all data underlying the findings in their manuscript fully available?

Reviewer #1: No

Reviewer #2: No

5. Is the manuscript presented in an intelligible fashion and written in standard English?

Reviewer #1: Yes

Reviewer #2: Yes

6. Review Comments to the Author

Reviewer #1: This paper is now very much improved. Thanks for taking the time to revise.Below are some points to be addressed before publication.

Line 18: Is this the correct order of steps?

Line 28: Reminders to do what?

Line 29: What are you referring to when you say “this”?

Line 32:Look into—change to explore?

Line 33- is perceive best? Experience? Yet again, not all will experience the benefits, so need to find a word here to match both.

Line 47-50 , hard to read and unsure of point being made.

Line 58- consider change seemed to appeared ?

Line 72-74- please consider using a comma or commas to make the sentence more readable.

Line 75- alternative word for seemed

Line 89-90: How is peer support a personal consideration?

Line 90- Please explain what you mean by novelty here- it is not the same “novelty” you refer to about the novelty of the desks in the paragraphs above and this may confuse readers

Line 99-102 it is still not really clear here whether you conducted this survey study too? Is it a precursor study to the one being reviewed here? You identified the sub groups from your own previous work? Or from someone else’s work? Although it is mentioned by the authors, I think the order that things are mentioned could be clearer.

Line 163-164: English here could be tightened up.

Line 169- what implications did the processing have? Were changes to the topic guide etc. made based on comments?

Line 174-178: Why one coder for some, 2 for others?

Line 181-185: Unclear please re write. Data analysis section is still hard to follow in places especially 181-185

Line 186-188; unclear of the point being made re saturation. So it was or wasn’t reached? For some themes only? I am unfamiliar with saturation being reported in this manner, What implications does this saturation/lack of saturation have in the discussion and interpretation of results?

Line 230- not sure if oyu can be from a nationality? Perhaps from a country is more correct.

Line 230-231how does biking to work impact of sitting at work?

Line 322: Consider “while standing”

Line 424: Consider using the term “role model”, but then be careful of the frequency of the term “role” in this section.

Line 557-558: How cold you do this?

Line 559- what does shared components mean

Reviewer #2: Thank you for attending to my comments. I have made a number of further comments that I think will strengthen your manuscript. The line numbers refer to the manuscript with tracked changes.

Abstract:

Line 17: Organization- UK or US spelling?

Line 20: thematic coding or thematic analysis? Please ensure terminology is consistent throughout the manuscript.

Line 24: the authors state that “staying alert and increasing focus” are facilitators for the use of standing desks. Would these factors not be “perceived benefits” rather than facilitators? In my mind a facilitator would be something like workplace culture or management support (i.e. something that makes it easier for an individual to participate in the behaviour), rather than a reason/motivation/ perceived benefit of the activity. If this is to be changed here, it will also need to be changed throughout the manuscript.

Introduction

Line 54- suggest changing this to “Recent studies exploring the use of”.

Line 56: Sentence starting “randomised controlled trials”: I realise the additions to this sentence have been due to a number of reviewer comments, however I think this sentence needs to be sense checked and perhaps split into two sentences. It is very long and difficult to read.

Line 76-83: Could these two sentences be condensed into one sentence to avoid repetition?

Method

Line 182- perhaps this reference will assist the authors in their discussion around data saturation

Clarke, V., & Braun, V. (2020). To saturate or not to saturate? Questioning data saturation as a useful concept for thematic analysis and sample-size rationales. Qualitative Research in Sport, Exercise and Health.

Line: 183-184: In my previous comments I asked for more detail about the “processing of comments and queries” here. While the authors have attempted to address this comment, I do not think enough detail has been provided. While the authors have now included that “minor changes were made”, I would prefer slightly more detail here. Could the authors provide a sentence here to summarise the changes that were made?

Line 188: Number 1-10 should be written out i.e. “Phase One...Phase Two”. This needs to be corrected throughout the manuscript.

Results

Line 210: Throughout the results section, where a quote is presented within a quote there are inconsistencies in the type of quotation marks used (e.g. differences between quotes presented on line 268 and line 495). I suggested checking all quotes throughout the manuscript to standardise the quotation marks used.

Line 211: Throughout the methods section the authors refer to “two main themes and one extra separate theme”, is there a reason this is not simply a third theme? Furthermore, the results are presented in a different order to the way the methods are presented. Methods are presented two main themes and then “extra theme”, whereas the results section starts with the “extra theme”. I suggest either reordering either the methods or results so that the themes are presented consistently throughout the paper.

Line 340: “Rationales behind putting the workstation up included to be at the same level as the visitor, shorter meetings and more intense conversations while being standing”. This sentence is quite long and difficult to follow. Could it be simplified? Also I am unsure what “shorter meetings” means? Does that mean that the standing desks can be used for short meetings, or the use of the standing desk shortens the length of a meeting and therefore this is a benefit? Also what does “more intense conversations while being standing mean”, what is an intense conversation? I think this description needs to be reconsidered.

Line 358: The use of the phrase “switch back” should be reconsidered throughout the manuscript as per my previous review comments as this is an example of colloquial language.

Line 367: Incorrect grammar here, rephrase “more easy” to “easier”.

Line 374: Should this read “Active alternatives to sedentary activities”? The ending of this sentence as it is currently written leaves me wondering “active alternatives to what”?

Line 393: Should this read “as a potential reason” rather than “as potential reason”

Line 436: I agree with the authors here that it is best to not change participant quotes as much as possible. To indicate that a change has been made to the participant quote here, I think the following quote should be written like this:

“it can be promptious [sic, a prompt] seeing a colleague with their desk up and so I think ‘I have not done this for a while’.”

Discussion

Line 516: Is there any other evidence to indicate that the use of standing desks improves concentration/ alertness which could be cited here?

Line 519: I suggest rephrasing this sentence to something like “users identified they preferred to use the standing desk when completing tasks which did not require high levels of concentration, such as sending emails”.

Line 544: The sentence beginning “Still, high-focus” is particularly long and difficult to follow. Could this sentence be simplified or split into two sentences?

7. PLOS authors have the option to publish the peer review history of their article (what does this mean?). If published, this will include your full peer review and any attached files.

Reviewer #1: Yes: Dr Aoife Stephenson

Reviewer #2: No

---

## [Author Response · Author response to Decision Letter 2]

6 May 2020

Please see the attached response to the reviewers.

---

## [Decision Letter · Decision Letter 3]

27 May 2020

PONE-D-20-01975R3

The user and non-user perspective: experiences of office workers with long-term access to sit-stand workstations

PLOS ONE

Dear Dr. Renaud,

Thank you for submitting your manuscript to PLOS ONE. After careful consideration, we feel that it has merit but does not fully meet PLOS ONE’s publication criteria as it currently stands. Therefore, we invite you to submit a revised version of the manuscript that addresses the points raised during the review process.

Please find comments from two expert reviewers below. As you can see, Reviewer 2 is largely happy with the manuscript and has only minor typographical amendments to suggest. Reviewer 1 still has concerns about aspects of the thematic analysis reporting, details of which can be found below. Please ensure these concerns are fully addressed in your revision. As the majority of other comments relate to syntax and grammatical errors, I support Reviewer's 1 suggestion of having the article thoroughly proof read by a native English speaker if possible. 

We look forward to receiving your revised manuscript.

Kind regards,

Kathryn L. Weston, PhD

Academic Editor

PLOS ONE

Reviewers' comments:

Reviewer's Responses to Questions

**Comments to the Author**

1. If the authors have adequately addressed your comments raised in a previous round of review and you feel that this manuscript is now acceptable for publication, you may indicate that here to bypass the “Comments to the Author” section, enter your conflict of interest statement in the “Confidential to Editor” section, and submit your "Accept" recommendation.

Reviewer #1: (No Response)

Reviewer #2: All comments have been addressed

2. Is the manuscript technically sound, and do the data support the conclusions?

Reviewer #1: Yes

Reviewer #2: Yes

3. Has the statistical analysis been performed appropriately and rigorously? 

Reviewer #1: N/A

Reviewer #2: N/A

4. Have the authors made all data underlying the findings in their manuscript fully available?

Reviewer #1: No

Reviewer #2: No

5. Is the manuscript presented in an intelligible fashion and written in standard English?

Reviewer #1: Yes

Reviewer #2: Yes

6. Review Comments to the Author

Reviewer #1: Thanks for the opportunity to re-review. This paper has improved significantly. My main concern is still revolving around the thematic analysis conducted and the reporting of this. I am still quite unclear how the themes were generated, the differences between main themes and subthemes for example. I find the figures explaining the trees don’t enhance the text as they are quite busy and hard to follow in places. I also have concerns around the reporting of saturation. Please see other papers where thematic analysis was used. This may help guide reporting. I feel this would also benefit from a thorough proof read from a native English speaker, ideally who has an understanding of scientific writing.

There are also some specific comments I feel need to be addressed as outlined below:

Line 40: suggest change “because” to “as”

Line 64: I feel by just saying that an increased reach could be yielded, you are underselling the impact. Please reconsider how else it may positively impact. Something along the lines of line 107-108

Line 67: Is the second comma required here?

Line70-72- word sitting used x 3, please re word for clarity

Line 76: Unsure what is meant by “not reducing long-term health risks”, please re word

Line 77-80: consider breaking in to 2 sentences for better readability

Line 91: Unsure who is being referred to in “as the users”- which users?

Line 92: Word perspective used x 2, consider another term eg, view. Also practitioners used x 2 in this sentence.

Line 92-93: Both practitioner and managers are mentioned, but only results for practitioners is explained in the end of the sentence? Does this mean the managers were not informed?

Line 100: Please check if this “n=” needs to be capitalised?

Line 103: Perhaps change to “This current study”

Line 104: Insight into?

Line 115: To keep the tense consistent, should this read had had, and not have had?

Line 117: unsure what small centred means here?

Line 125: Since user is pronounced as starting with a consonant "y" sound, the article a is appropriate, and an is not.

Line 145: consider changing group interviews assisted to “group interviews, assisted”

Line 146:How many were familiar?

Line 163-164: This warm-up exercise for participants was intended to become familiar with the

topic and the group. This sentence does not read well. Perhaps something like “The purpose of the warm-up exercise was to allow participants become familiar with the topic and the group”.

Line 166-169: What was the purpose of doing this if changes were not made? I am struggling to see the relevance of this section, please clarify.

Line 183-187: I cannot understand how these themes were generated, please clarify. Confused re extra and main themes. Because I don’t fully grasp this section, interpreting the results is difficult

Line 183-191: Please consider revising this. Please read other papers where thematic analysis is used and how they report this. Saturation section also needs to be reworked.

Line 205: Table?

Line 206: to participate, consider changing to “in participating”

Line 207:Were the dates/times pre scheduled , or why may some have been unavailable?

Line 209: had long working experience, please reword to something like “long term employees”

Line 211: indicated to use, please change to something like “indicated using it”

Line 221: Capitalise “figure”? Please check is this is required and if so change “table” and “figure” in text throughout.

Line 255-256: Perhaps break into 2 sentences or rephrase- hard to follow

Line 257-259: any reason for this? Why some had a pattern and some preferred to use randomly?

Line 274: practical issues of the workplace- should this read “practical workplace issues”?

Line 274-7: Please reword this, or at least use better punctuation to ensure the reader doesn’t get confused by the “” s and theme names etc

283-285: Please re phrase or break into 2 sentences for clarity and readability

338-9: Should this last “and” be “or”?

416-7: Larger desk surfaces to adjust to standing height- please clarify what is meant here

424-6: Please re phrase or break into 2 sentences for clarity and readability

Line 439: Within the theme individual adjustments, If this read Within the theme “Individual Adjustments”,, it may help readability, please change where this occurs with other themes throughout paper.

442: What is a facility room?

444: Please consider changing comfortable to, to “comfortable for”

454: However, also other emphasis were made during these group interviews. This doesn’t make sense to me. Please rephrase

553-4: in which the organisational culture promotes health behaviour- should this be behaviours? Or what behaviour?

581: consider changing personal room to private office?

Reviewer #2: Thank you for updating the manuscript based on my previous comments. The manuscript is now much improved and will make an excellent contribution to the evidence base for sedentary behaviour interventions in the workplace. I have noted a few minor typographical errors below, however I do not feel the need to review this paper again, when these changes have been made, as they are very minor.

I suggest changing wording of the aim of the paper from “gain insight” to “explore”. This will need to be updated on lines 13, 103, 106 and 590.

Line 194- full stop at the start of the sentence needs to be removed

Line 108- change = to ± for standard deviation

Line 237- should “prove” be “proof”?

Line 472- formatting of quote within a quote needs to be corrected

7. PLOS authors have the option to publish the peer review history of their article (what does this mean?). If published, this will include your full peer review and any attached files.

Reviewer #1: No

Reviewer #2: No

---

## [Author Response · Author response to Decision Letter 3]

18 Jun 2020

Please see our letter to the editor and response to the reviewers.

---

## [Decision Letter · Decision Letter 4]

10 Jul 2020

The user and non-user perspective: experiences of office workers with long-term access to sit-stand workstations

PONE-D-20-01975R4

Dear Dr. Renaud,

We’re pleased to inform you that your manuscript has been judged scientifically suitable for publication and will be formally accepted for publication once it meets all outstanding technical requirements.

Kind regards,

Kathryn L. Weston, PhD

Academic Editor

PLOS ONE

Additional Editor Comments (optional):

Please note a few minor suggestions from the Reviewer below. Please ensure these are addressed ahead of the final publication. 

Reviewers' comments:

Reviewer's Responses to Questions

**Comments to the Author**

1. If the authors have adequately addressed your comments raised in a previous round of review and you feel that this manuscript is now acceptable for publication, you may indicate that here to bypass the “Comments to the Author” section, enter your conflict of interest statement in the “Confidential to Editor” section, and submit your "Accept" recommendation.

Reviewer #1: All comments have been addressed

2. Is the manuscript technically sound, and do the data support the conclusions?

Reviewer #1: Yes

3. Has the statistical analysis been performed appropriately and rigorously? 

Reviewer #1: N/A

4. Have the authors made all data underlying the findings in their manuscript fully available?

Reviewer #1: No

5. Is the manuscript presented in an intelligible fashion and written in standard English?

Reviewer #1: Yes

6. Review Comments to the Author

Reviewer #1: I am happy to accept this for publication under the assumption the following comments will be read and addressed. These are minor issues (mainly grammar).

Line 23: Check tense “experiencing”

Line 31: Please consider changing “would” to something less definite

Line 65: How does increasing quality thus lead to increasing the reach?

Line 66/67: Check tenses here “chose” is past tense and do is present.

Line 68: Is “must” the best term here? There are other ways perhaps to look into it-other than qualitative?

Line 145: US or UK spelling?

Line 185: Why are phase 4 and 5 grouped together?

Line 275/276: Are there missing quotation marks here around the theme name?

Line 392: Are there missing quotation marks here around the theme name?

Line 589: Are there missing quotation marks here around the theme name? Check this throughout document.

Does the lack of saturation need to be discussed in Limitations section?

7. PLOS authors have the option to publish the peer review history of their article (what does this mean?). If published, this will include your full peer review and any attached files.

Reviewer #1: No

---

## [Editor Report · Acceptance letter]

15 Jul 2020

PONE-D-20-01975R4 

The user and non-user perspective: experiences of office workers with long-term access to sit-stand workstations 

Dear Dr. Renaud:

I'm pleased to inform you that your manuscript has been deemed suitable for publication in PLOS ONE. Congratulations! Your manuscript is now with our production department. 

Kind regards, 

on behalf of

Dr. Kathryn L. Weston 

Academic Editor

PLOS ONE